# How to estimate health service coverage in 58 districts of Benin with no survey data: Using hybrid estimation to fill the gaps

**Alex Ocampo**[1], **Joseph J. Valadez**[2]*, **Bethany Hedt-Gauthier**[1,3], **Marcello Pagano**[1]

**1** Department of Biostatistics, Harvard T. H. Chan School of Public Health, Boston, MA, United States of America, **2** Department of International Public Health, Liverpool School of Tropical Medicine, Liverpool, United Kingdom, **3** Department of Global Health and Social Medicine, Harvard Medical School, Boston, MA, United States of America

* joseph.valadez@lstmed.ac.uk

**Data Availability Statement:** The probability survey data has been uploaded to our online archive. Interested readers can download our data

## Abstract

The global movement to use routine information for managing health systems to achieve the Sustainable Development Goals, relies on administrative data which have inherent biases when used to estimate coverage with health services. Health policies and interventions planned with incorrect information can have detrimental impacts on communities. Statistical inferences using administrative data can be improved when they are combined with random probability survey data. Sometimes, survey data are only available for some districts. We present new methods for extending combined estimation techniques to all districts by combining additional data sources. Our study uses data from a probability survey (n = 1786) conducted during 2015 in 19 of Benin's 77 communes and administrative count data from all of them for a national immunization day (n = 2,792,803). Communes are equivalent to districts. We extend combined-data estimation from 19 to 77 communes by estimating denominators using the survey data and then building a statistical model using population estimates from different sources to estimate denominators in adjacent districts. By dividing administrative numerators by the model-estimated denominators we obtain extrapolated hybrid prevalence estimates. Framing the problem in the Bayesian paradigm guarantees estimated prevalence rates fall within the appropriate ranges and conveniently incorporates a sensitivity analysis. Our new methodology, estimated Benin's polio vaccination rates for 77 communes. We leveraged probability survey data from 19 communes to formulate estimates for the 58 communes with administrative data alone; polio vaccination coverage estimates in the 58 communes decreased to ranges consistent with those from the probability surveys (87%, standard deviation = 0.09) and more credible than the administrative estimates. Combining probability survey and administrative data can be extended beyond the districts in which both are collected to estimate coverage in an entire catchment area. These more accurate results will better inform health policy-making and intervention planning to reduce waste and improve health in communities.

set used in the paper at: https://archive.lstmed.ac.
uk/20297/.

**Funding:** This work was supported, in whole or in
part, by the Bill & Melinda Gates Foundation
[Investment ID OPP1142889]. Under the grant
conditions of the Foundation, a Creative Commons
Attribution 4.0 Generic License has already been
assigned to the Author Accepted Manuscript
version that might arise from this submission. This
work was also funded by UNICEF New York
[Contract Number: 43114258]. Alex Ocampo was
supported by NIH-5T32AI00735. The funder played
no role in this research.

**Competing interests:** The authors have no
competing interests to declare.

# Background

Disparities in the availability of resources and unequal access to them exist across all aspects of the global healthcare landscape. Access to reliable health care data to manage the health system is no exception [1–3]. Achieving Sustainable Development Goals (SDG) and Universal Health Care (UHC) is now a global priority (United Nations General Assembly Resolution A/RES/70/1). However, managing health services so all individuals and communities receive the care they need without suffering financial hardship requires accurate information about a community's access to a full spectrum of quality services [4]. Otherwise, evidence-based programming cannot take place [2]. The continuing lack of reliable health data in low and middle-income countries is well documented [5–7]. Now more than ever, the global community recognizes the essential need for quality data from all countries, so that impacts of current health policies can be assessed by each country, but also to guide improvements of current health practices and healthcare interventions in local communities.

One promising trend in developing countries is the continued adoption of data entry into electronic systems during routine health care services [8], although most health information systems are paper-and-pencil based. Both systems contain counts of interventions such as the number of infants who have received a childhood vaccination; however, extrapolating this information to estimate the proportion of a population receiving them is often questionable without making adjustments with alternative data sources [9]. Even if the number of people in a particular catchment area receiving an intervention were tallied correctly (the numerator), the number of eligible people in a particular catchment area (the denominator) is needed to accurately estimate prevalence. Lack of accurate or timely census data in many areas can limit our ability to accurately calculate prevalences using these numerators. Such deficiencies lead to inaccurate estimates, misguided health system management decisions, wasted resources and preventable illnesses.

A viable strategy to approximate an accurate measure of health indicators is to utilize all available data. Optimal statistical methods can synthesize knowledge from different sources, taking advantage of their disparate strengths. One approach to form prevalence estimates is to combine administrative data with random surveys and leverage both of their statistical properties [9,10]. Combining administrative data with data from random samples is often more efficient than random samples alone–and thereby less expensive for the same precision–and can correct biases inherent to using administrative data alone. Probability surveys are representative of the population because they employ randomization to sample from either the whole population or within well-defined strata [11].

With this in mind, Jeffery et al in 2018 advanced techniques for combining data in an approach called "hybrid prevalence estimation" which combines large-scale administrative data from a Child Health Day with probability household survey data collected using Lot Quality Assurance Sampling (LQAS) [9]. LQAS has been used widely throughout the developing world to assess disease prevalence, immunization coverage, and behavioural health indicators [12–14]. The statistics from both administrative and LQAS survey data are combined using weights estimated using the relative efficiency of the statistics to formulate one final hybrid estimate and associated standard error. To demonstrate the utility of their approach, they estimated vitamin A supplementation (VAS) and polio vaccination rates in children from Benin. We utilize the same data set to demonstrate an extension of their approach to situations where the survey data is not available at all locations.

While the hybrid prevalence approach provides estimates in districts in which random samples have been conducted, it is not always feasible to perform probability surveys in all administrative areas. In districts without probability surveys, there is only count data on the number

of children receiving vaccinations and VAS. Dividing this count by the number of children in the population, i.e. the "denominator", could yield the proportion of children vaccinated. However, since the last census in Benin was conducted in 2013, this denominator can only be estimated. Previously published estimates of this denominator have either used the numerator from the Child Health Day of six-months ago [9] or the extrapolation of old census data adjusting for the population growth rate [15]. The former choice assumed all children 12–59 months were contacted via a door-to-door campaign and that the current cohort of children is the same size. Both approaches are sometimes questionable, especially when they lead to coverage rates that are larger than 100%, as was the case in [9]. To date, there is no hybrid estimator solution for areas with administrative data, but without a corresponding random sample with which to anneal administrative data. To address this limitation, we propose a solution that leverages regions with both probability surveys and administrative data to posit a model that extrapolates estimates to an entire national catchment area without conducting additional probability surveys.

## Data overview

To illustrate our proposed approach for estimating intervention coverage rates for each administrative area in a given country, we consider data on polio vaccinations and VAS given to children under five-years of age in Benin during a Child Health Day [16]. Our goal is to estimate coverage rates, which is the percentage of children receiving these interventions in each of the 77 communes which constitute the geographical administrative units in Benin. They are comparable to districts in several other countries (e.g., Kenya, Uganda). Although these data have been described in detail elsewhere [9], we give a brief overview in this section to illustrate the fundamental issues and the need for new methodology. We leverage two types of data available for communes in Benin: 1) small sample probability surveys estimating the percentage of children receiving the intervention and 2) large administrative data counting the number of children receiving a polio vaccination and VAS during the Child Health Day (analogous to a National Immunization Day). While administrative tallies are available for each of the 77 communes in Benin, probability surveys are only available for 19 communes. The research question we address is: using the above two sources of data, is it possible to estimate prevalences in the 58 communes without probability surveys?

## The probability surveys

Stratified random probability surveys were conducted in 19 of Benin's 77 communes between November 16–20, 2015. Selected communes were divided into administrative units called supervision areas (SA). Each commune had five SAs, in each of which 19 children 12–59 months of age were surveyed randomly, for a total of 95 children surveyed per commune. The only exception was the commune of Ouinhi, which had four SAs, resulting in only 76 children surveyed. The total sample size was 1786 respondents. The sampling scheme has been published previously in detail [12,17,18]. Structured questionnaires were used to measure multiple health indicators, including both polio vaccination and VAS.

Benin aggregates communes into departments. The 19 sampled communes came from three departments of Benin: nine communes from the departments of Alibori and Atacora in the Northeast, all nine communes of the central department of Zou, and one commune near the coastal capital. Since this set of communes was not chosen by random chance, those communes with missing probability surveys cannot be classified as missing completely at random [19].

### Administrative data

The Ministry of Health (MOH) in Benin collects data on the administration of VAS and polio vaccination to eligible children during a Child Health Day [16,20], which recurs every six months. We focus on data from 2015. Vaccination teams record on tally sheets the numbers of children they served in each of Benin's 77 communes. These tallies are useful as the resulting number of children served during these campaigns should be indicative of VAS and polio vaccination rates in a given commune. If so, these count data could be used as a numerator and then combined with a suitable estimate of the denominator to yield prevalence estimates.

## Methods

### Estimating denominators

**Ratio estimation of denominators.**   We used the probability survey data to estimate denominators using the relationship:

$$p = \frac{n}{d}$$

where for each commune, $p$ was the probability of polio vaccination, $n$ the number of children recorded in the tally sheets as being vaccinated, and $d$ the number of children residing in the area where the vaccination took place. When estimating $n$, we used the number of children recorded in the tally sheets as vaccinated in all communes; however, we had no estimates of $d$ due to the lack of 2015 census data. Because in some communes we had a reliable estimate of $p$ from the probability surveys, we were able to estimate $d$ for these communes by transforming the earlier relatonship:

$$\hat{d} = \frac{n}{\hat{p}}$$

Here, $\hat{p}$ was the probability survey estimate, which, in our example in Benin, were available for 19 communes. We then calculated $\hat{d}$ for these 19 communes, and subsequently used them in conjunction with the denominators from the tally sheets from the Child Health Day of 6-months earlier to estimate the remaining communes in Benin.

**Estimating denominators with multiple health indicators.**   In many cases multiple indicators are measured by both probability survey and administrative data. For example, in Benin we present data on two health indicators, polio vaccination and VAS coverage. While the coverage rates for these indicators do differ across communes, the denominator we are estimating in a commune, e.g. the number of children age 12–59 months, is the same for both indicators. Therefore, we can combine data across health indicators to improve estimates of the denominators. We propose doing so using an inverse variance weighted denominator estimate. Our original probability survey estimates $\hat{p}_1$ and $\hat{p}_2$ come with estimated variances. When these estimators are combined with numerators to form the estimators of the denominators $\hat{d}_1$ and $\hat{d}_2$, we can use the delta method to obtain valid variance estimates for the denominators. Consider the weights $w_k = 1/\hat{\sigma}^2_{d_k}$, which are higher when the variances are lower, then the estimator is:

$$\hat{d} = \frac{w_1\hat{d}_1 + w_2\hat{d}_2}{w_1 + w_2}$$

### Predicting denominators

**Initial approach.**   We estimate denominators in the 58 communes that do not have probability surveys by first establishing a regression relationship within the 19 communes that do have probability surveys, and then use this model predictively for the remaining 58 communes.

Consider fitting the following regression model:

$$\hat{d}_j = X_j\beta + \varepsilon_j \tag{1}$$

Where $\hat{d}_j$ are the estimated denominators for the $j = 1,\ldots,19$ communes with probability surveys, $X_j$ is a matrix of covariates, and $\varepsilon_j$ is the residual term with mean 0 and finite variance. This yields estimates, $\hat{\beta}$. Since the $X_j$ are available for all communes, we can then apply the formula,

$$\hat{d}_j = X_j\hat{\beta}$$

for $j = 20,\ldots,77$, the communes without probability surveys. Variance estimates can be obtained in the standard way. Then by combining these predicted denominators with the known numerators we can get estimates of the $\hat{p}_j$ for communes $j = 20,\ldots,77$.

**Bayesian approach.**   One drawback of the approach above is that the linear model postulated for the denominator above does not guarantee that estimated denominators be larger than the known numerators. As a consequence, it is possible to obtain estimates of $\hat{p}_j > 1$; which we know would be incorrect. If this is the case, we could use the ad hoc estimate, $\hat{p}_j = 1$, which has no other justification than that it does not exceed one. The proposed Bayesian approach begins by postulating that the denominators $d_j$ for $j = 20,\ldots,77$, are random variables with probability distributions centered at the previously predicted estimates $\hat{d}_j$ with variances consistent with an out of sample prediction interval ($\sigma^2_{\hat{d}_j}$). For example, we could use the normal model, with the $d_j$ independent:

$$d_j \sim N(\hat{d}_j, \sigma^2_{\hat{d}_j})$$

Above $d_j$ represented the denominators for the 58 communes without probability surveys. This distribution quantified our uncertainty about the estimated denominator and also led to the distribution of the coverage estimates $p_j$. We then refined our model by truncating denominator estimates, $d_j$ to values above $n_j$, to ensure that estimates of $p_j$ were less than 1. We described this as "truncation by reality" since we knew the denominator cannot be less than the number of children vaccinated in a particular commune (i.e. vaccination rates could not have been higher than 100%). More specific details of the Bayesian strategy can be found in S1 Text. Lastly, the Bayesian framework allowed us to easily conduct a sensitivity analysis by shifting the posterior predictive distributions. We included details on the sensitivity analysis procedures in S2 Text.

## Application to Benin data

**Analysis overview.**   In this section we present the hybrid extrapolation approach's utility in estimating polio vaccination coverage among children age 12–59 months at the commune level in Benin in 2015. We wish to leverage the survey data from 19 communes to the 58 communes for which we only have administrative data.

For commune factors we used population estimates for each commune from the 2013 census [21] as well as vaccination tallies from 6-months earlier. Both of these variables are strongly associated with our estimated denominators (Fig 1).

Using these two commune factors, we fit the regression model as defined in the methods section, using the estimate of the denominator as our outcome variable. We fit this model on the 19 communes that had probability surveys for which we have an unbiased estimate of the

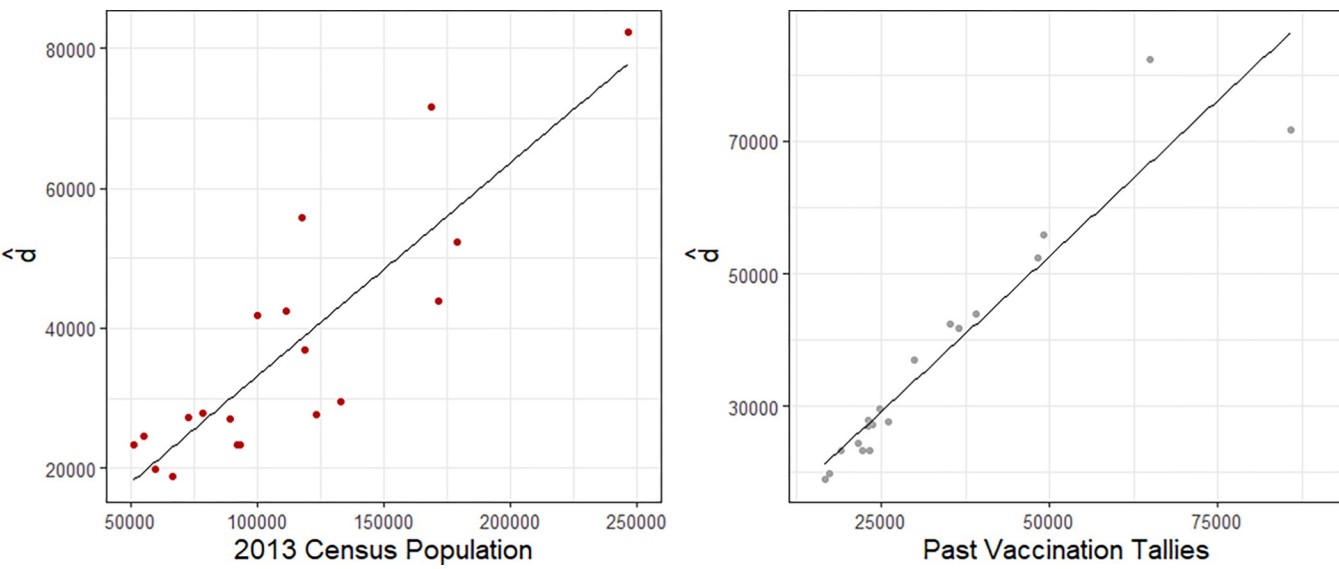

**Fig 1. Census data (2013) & past vaccination tallies from Child Health Day 6-months earlier vs estimated denominators.**

denominator. This model fit the data well and had an adjusted $R^2$ = 0.9452. Using this model, we predict denominators for the 58 communes not having probability survey data. We then divide the tally sheet numerators in those communes by the model predicted denominators to form hybrid prediction estimates.

## Ethics approval and consent to participate

The Ethical Committees of UNICEF NY and UNICEF Benin, and the Liverpool School of Tropical Medicine Research Ethics Committee approved the protocol (12.10), study instruments and consent procedures for the LQAS surveys. All interviewees were read an approved informed consent statement indicating the purpose of the survey, the expected amount of time the interview would take, and that participation was voluntary; they were then invited to ask questions about the survey before giving their written informed consent.

## Results

The estimated polio vaccination rates and corresponding 95% confidence interval for children ages 12–59 months in each commune using the direct regression estimates are shown in (Fig 2). These include both the baseline hybrid estimators from the 19 communes (grey dots) and the extrapolated hybrid estimates (red dots). The obvious issue is that 6 out of 77 of the model estimated commune vaccination rates are above one. However, this is an improvement compared to previously published attempts to estimate these denominators with administrative data alone, which led to 58 of 77 communes with coverage estimates above one (black dots).

A number of communes have noticeably large confidence intervals in the model results; this is because the covariate values for these communes are small relative to the covariates of the surveyed communes used to fit the predictive model. The largest of these confidence intervals belong to communes that have populations less than 60 000 people, which falls below the inter-quartile range of commune population sizes–i.e. these predictions are extrapolations.

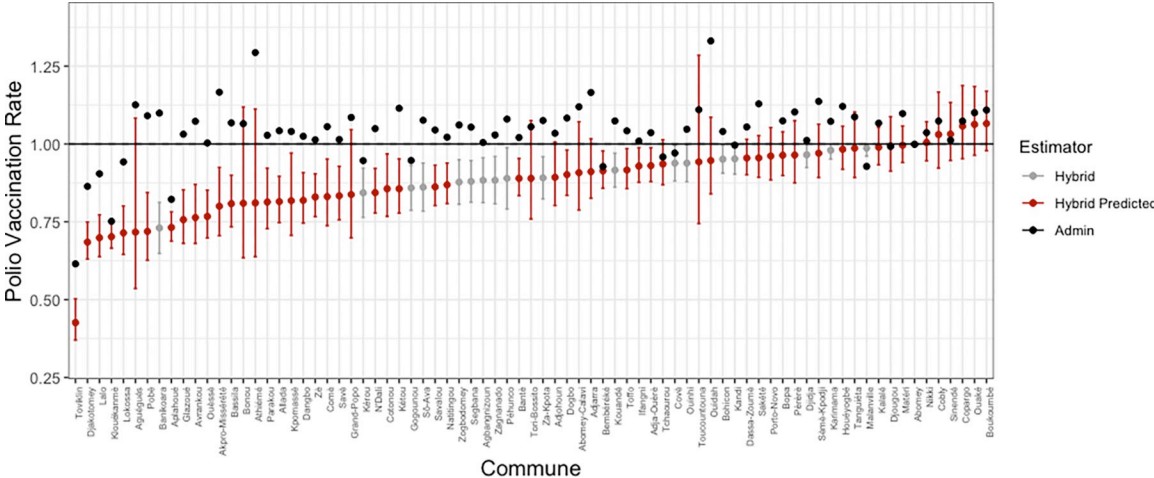

**Fig 2. Frequentist estimates of polio vaccination rates by commune.**

## Bayesian model

The Bayesian formulation solves the estimation problem. The resulting posterior medians and 95% credible intervals for the same polio vaccination rates in Fig 2 are shown in Fig 3. This figure confirms that it is impossible to obtain either point estimates or intervals above one using the Bayesian approach.

We conducted a sensitivity analysis to test the robustness of our inferences. Specifically, we were interested in 5% and 10% bias in the administrative tally sheet data. Lowering the administrative numerators by 10% lowers the coverage estimates by 6%, on average. Increasing the administrative numerators by 10%, increases the coverage estimates by 4%, on average. The upward bias has a smaller effect because our probabilities are bounded above by 1. See Additional file 2 for more details on the approach. The resulting effects on the commune level can be seen in Fig 4.

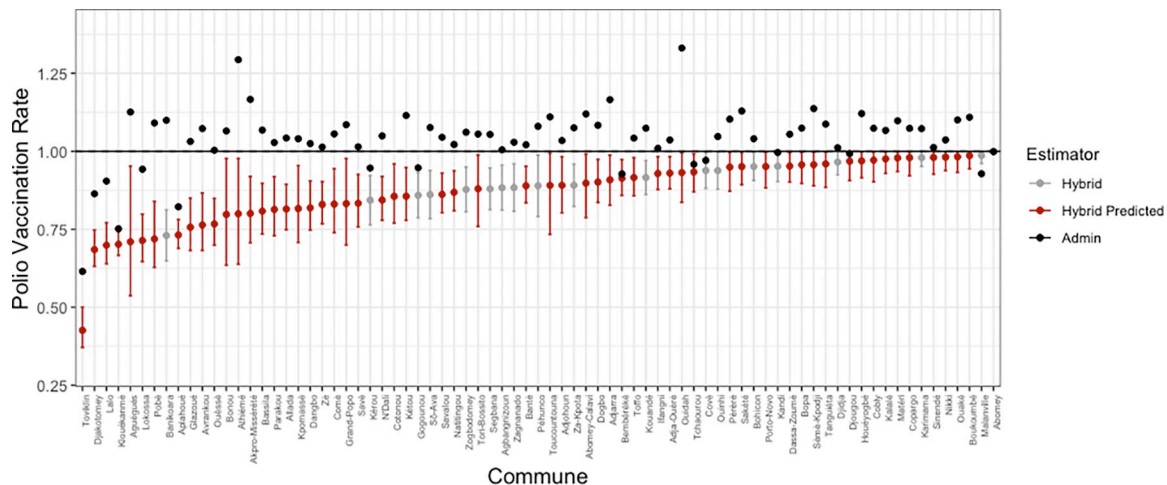

**Fig 3. Bayesian estimates of polio vaccination rates by commune.**

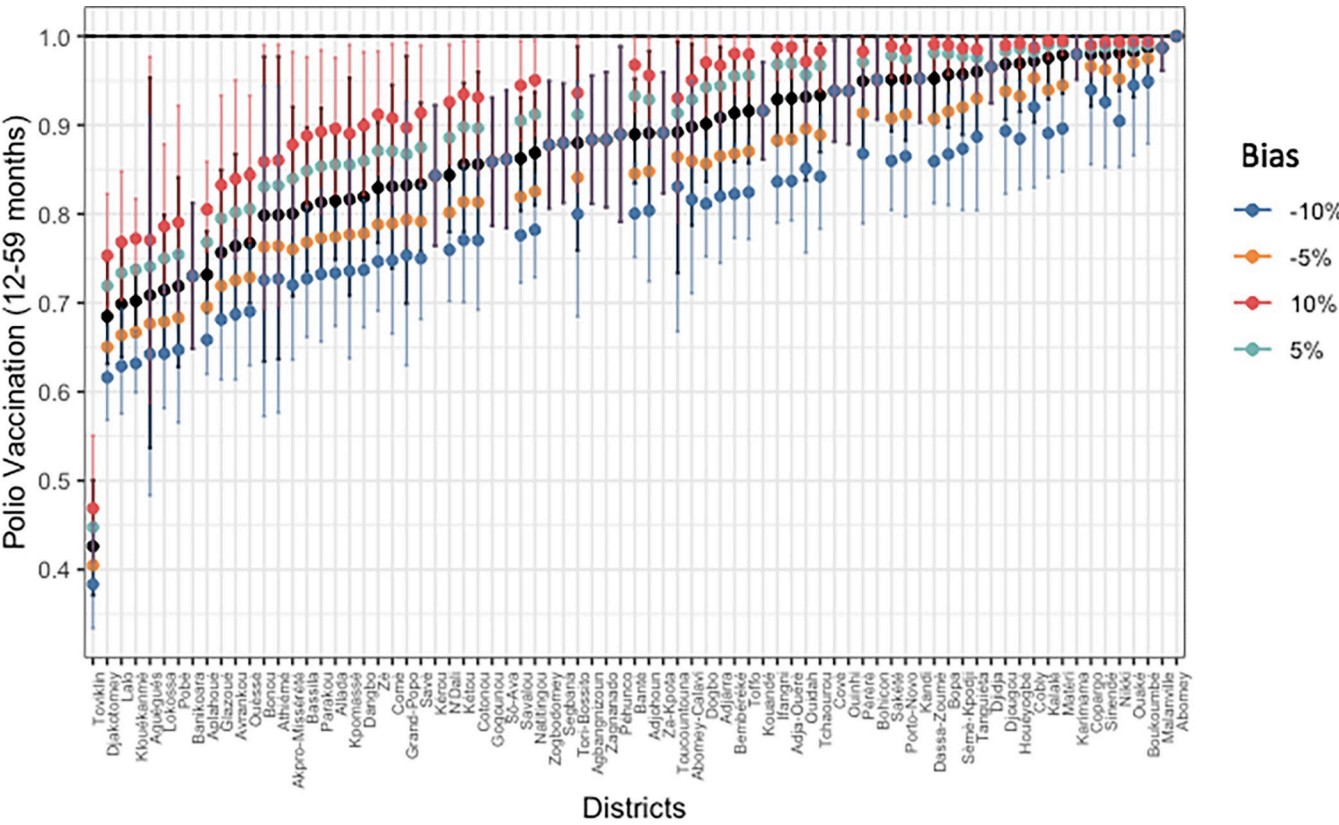

**Fig 4. Sensitivity analysis of polio vaccination rates by commune.**

Results for other indicators and age groups are presented in S3 Text. Lastly, a negative binomial model was further considered for the denominators in place of the normal model. The results using this alternative distribution are presented in S4 Text.

## Discussion

We provide coverage estimates for all communes in Benin by combining administrative tally sheet data from a child immunization campaign with repurposed random probability survey data. We accomplish this with a ratio-type estimator that we model to extrapolate estimates to communes not covered by the probability survey data, but related to the covered communes by the fact that they all share a common countrywide administrative system in Benin. In addition, we describe a Bayesian approach to the problem, that ensures point estimates and intervals which fall within permissible ranges (0–100%) making them useful and credible.

Our methodology can profit public health practitioners looking to leverage both administrative and probability survey data. Especially in the digital age, administrative data are inexpensive when routinely collected as a part of health services. These data-sets are often large and cover a wide geographical range, such as the entire nation of Benin. Despite these advantages, administrative data are convenience samples, and not necessarily representative of the population of interest. Probability survey data accurately estimate population prevalences when sampling is done appropriately. The downside is that, for economic reasons, sample sizes may be small and limited to certain geographical regions. For instance, the probability survey data used here only contained estimates for 19 out of 77 communes in Benin, and

roughly 100 households were sampled in each commune. Our methodology takes the best from both sources of data, and combines them using a statistically principled approach. We demonstrate this principled approach for extrapolating survey estimates to districts with only convenience administrative data being available. Through extrapolation we allow surveyors to produce nationwide vaccination rate estimates while sparing them from the herculean task of conducting nationwide surveys.

Our approach and analysis are subject to limitations. The approach is a technical one, and will require statistical experience, perhaps even at the district level to analyse and communicate results. However, history has shown that complex quality control statistics can be transformed into user friendly tools for local users [14,22,23] While our regression model fits the data well, we only utilized two covariates of district population sizes, and assumed a simple linear form. Analyses could benefit from more complex models that take into account a larger and more diverse set of covariates of health service access (e.g., road density or mothers attending ante-natal care). A sensitivity analysis conducted using only the census data as a sole covariate resulted in less accurate estimates (S5 Text), making the case for incorporating even more covariate information. In addition, some of the confidence intervals of our estimates are wide for certain communes. Specifically, these were communes with very small populations relative to the median commune size, having reported less than 60,000 inhabitants in the 2013 census. The median population size for communes in our model fit was 100,197 (Inter-Quartile Range [IQR]: 75,383–128,180). To avoid such extrapolations, we recommend the probability surveys purposefully sample some areas with population sizes outside of the IQR. Additionally, a great deal of confidence is placed in the administrative numerators in forming initial denominator estimates for which our model was fit. Systemic errors have been found when using administrative errors to generate public health statistics [22]. Administrative data, such as the number of people vaccinated in a given year, are often collected in the field and may be biased due to incomplete collection, inaccurate entry, or other errors. If the administrative data are biased, then the foundational assumption of the approach is violated, which will induce bias in the extrapolation. This limitation highlights the importance of a sensitivity analysis to understand how changes in the administrative data could affect scientific conclusions downstream. However, bias due to the incompleteness or inaccurate data entry into records can be reduced by Data Quality Audits, Routine Data Quality Assessments, the use of electronic data entry; these approaches have been shown to improve data accuracy in lower and middle income countries [24–27]. Lastly, the vaccination prevalences tended to be high in the districts of Benin sampled. However, if prevalence probabilities happen to be small for a given survey in a different context (e.g. prevalence of a rare disease), this may lead to higher variability in the final estimates.

We recommend future research explore the utility of hybrid extrapolation in both theory and practice. Firstly, statistical properties of these estimators should be more robustly evaluated in simulation studies. More practically, public health practitioners with access to both administrative count data and survey data can attempt the extrapolation and validate extrapolated estimates with subject matter experts to ensure that prevalence estimates are sensible. One impactful area of opportunity is in the context of COVID-19, where administrative confirmed case counts are often available for many districts, but random antibody testing surveys are less often conducted.

Estimates published previously resulted in 75% of the polio coverage rates for children aged 12–59 months to be above 100%. Thus, formulating a solution to the large coverage rates using the Bayesian perspective seems appropriate, as it forces our coverage estimates below 100%. It also enables a simple and flexible approach to a sensitivity analysis.

## Conclusion

This approach should motivate researchers to further consider the merits of combining administrative data with results from random probability survey data to produce more accurate data to inform policy-making and planning of healthcare interventions for communities. Our work demonstrates that not only could our innovation have implications for geographic regions with both sources of data available, but also adjacent regions can benefit from annealing data sources as well. Improving key indicator estimates by combining data sources provides a more comprehensive view of the health system's coverage and can efficiently contribute to advancing Universal Health Care. It can also lead to more accurate assessments of the current impacts of health policies generate debate for their improvement.

## Supporting information

**S1 Text. Details of the Bayesian approach and truncation of the posterior predictive.** Description: Further details outlining the Bayesian approach described in the methodology section.
(DOCX)

**S2 Text. Title of data: Bayesian sensitivity analysis for predicted denominators.** Description: The sensitivity analysis performed assuming a varying numerator in the communes. An outline of the approach for the sensitivity analysis and observed changes in the prevalence estimates assuming different numerators are presented.
(DOCX)

**S3 Text. Results for other health indicators and populations in Benin.** Description: Results for the three other health indicators and populations using the same methodology presented in the main text are included.
(DOCX)

**S4 Text. Results for a negative binomial bayesian model.** Description: Results using the negative binomial distribution.
(DOCX)

**S5 Text. Results for estimating denominators using only population data.** Description: This sensitivity analysis reveals that reducing the predicted denominators to one covariate decreases accuracy of the "hybrid predicted" estimates (i.e. wider confidence intervals).
(DOCX)

**S6 Text. Inclusivity questionnaire.** Description: This file includes reduces responses to PLoS Global Health inclusivity in global research questionnaire.
(DOCX)

## Acknowledgments

The authors want to acknowledge the Benin Ministry of Health as well as the healthcare workers and surveyors who collected the data. Without them this research would not have been possible.

## Author Contributions

**Conceptualization:** Alex Ocampo, Joseph J. Valadez, Bethany Hedt-Gauthier, Marcello Pagano.

**Data curation:** Alex Ocampo.

**Formal analysis:** Alex Ocampo, Joseph J. Valadez, Marcello Pagano.

**Funding acquisition:** Joseph J. Valadez, Marcello Pagano.

**Investigation:** Alex Ocampo, Joseph J. Valadez, Bethany Hedt-Gauthier, Marcello Pagano.

**Methodology:** Alex Ocampo, Joseph J. Valadez, Bethany Hedt-Gauthier, Marcello Pagano.

**Project administration:** Joseph J. Valadez, Marcello Pagano.

**Software:** Alex Ocampo.

**Supervision:** Joseph J. Valadez, Bethany Hedt-Gauthier, Marcello Pagano.

**Validation:** Marcello Pagano.

**Visualization:** Alex Ocampo, Joseph J. Valadez, Marcello Pagano.

**Writing – original draft:** Alex Ocampo, Joseph J. Valadez, Bethany Hedt-Gauthier, Marcello Pagano.

**Writing – review & editing:** Alex Ocampo, Joseph J. Valadez, Bethany Hedt-Gauthier, Marcello Pagano.

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
