## [Decision Letter · Decision Letter 0]

10 Nov 2021

PGPH-D-21-00820

How to estimate health service coverage in 58 districts of Benin with no survey data: using hybrid estimation to fill the gaps

Dear Dr. Valadez,

Thank you for submitting your manuscript to PLOS Global Public Health. After careful consideration, we feel that it has merit but does not fully meet PLOS Global Public Health’s publication criteria as it currently stands. Therefore, we invite you to submit a revised version of the manuscript that addresses the points raised during the review process.

We look forward to receiving your revised manuscript.

Kind regards,

Karen A. Grépin, Ph.D.

Academic Editor

Journal Requirements:

2. In the online submission form, you indicated that "The probability survey datasets generated and/or analysed during the current study are available from the corresponding author on reasonable request. The administrative data is owned by UNICEF Benin from whom a request should be made."

3. Please provide us with a direct link to the base layer of the map used in figure 1 and ensure this location is also included in the figure legend. 

Please note that, because all PLOS articles are published under a CC BY license (creativecommons.org/licenses/by/4.0/), we cannot publish proprietary maps such as Google Maps, Mapquest or other copyrighted maps. If your map was obtained from a copyrighted source please amend the figure so that the base map used is from an openly available source.

Please note that only the following CC BY licences are compatible with PLOS licence: CC BY 4.0, CC BY 2.0  and CC BY 3.0, meanwhile such licences as CC BY-ND 3.0 and others are not compatible due to additional restrictions. If you are unsure whether you can use a map or not, please do reach out and we will be able to help you. 

The following websites are good examples of where you can source open access or public domain maps:

Additional Editor Comments (if provided):

We thank you for submitting your manuscript to PLOS Global Public Health. I have reviewed the feedback from both reviewers, and have also reviewed the article myself.

Both reviewers agreed that the paper addresses a very important issue in global health measurement: the ability to predict denominators to estimate population coverage levels from administrative data. I strongly agree with the reviewers on this point.

However, both reviewers have also raised some important statistical concerns about the methods used and therefore I would encourage you to consider their feedback in your revision of your manuscript.

I would also add another concern that was not directly raised by the reviewers, which is that the authors frame the introduction about the need to make better use of routine health information system data (also referred to as "recurrent" a term I have not seen widely used and that I find confusing), yet the data they use comes from campaign data, which while administrative is not necessarily "routine" by many definitions. Could the authors comments on how this many affect the generalisability of the findings?

Also, stylistically, could the authors avoid calling their approach "innovative" and "novel" as these are highly subjective terms? I think "new" is sufficient.

Reviewers' comments:

Reviewer's Responses to Questions

**Comments to the Author**

1. Does this manuscript meet PLOS Global Public Health’s publication criteria? Is the manuscript technically sound, and do the data support the conclusions? The manuscript must describe methodologically and ethically rigorous research with conclusions that are appropriately drawn based on the data presented.

Reviewer #1: Partly

Reviewer #2: Partly

2. Has the statistical analysis been performed appropriately and rigorously?

Reviewer #1: Yes

Reviewer #2: Yes

3. Have the authors made all data underlying the findings in their manuscript fully available (please refer to the Data Availability Statement at the start of the manuscript PDF file)?

Reviewer #1: No

Reviewer #2: Yes

4. Is the manuscript presented in an intelligible fashion and written in standard English?

Reviewer #1: Yes

Reviewer #2: Yes

5. Review Comments to the Author

Reviewer #1: The authors developed a hybrid method to estimate the denominators in all catchment areas by combining administrative data with random probability survey data available in only certain parts of the population. A Bayesian approach with Normal model was used. This denominator-generating model was also truncated to be above the observed number of vaccinations given (the numerators) so that the resultant vaccine coverage would never exceed 1. The authors also conducted a sensitivity analysis and an application to estimate the commune-level polio vaccination coverages among children aged between 12-59 months in Benin, where only 19 out of the 58 communes had survey data.

The issue that the manuscript aims to address is a very common and important question in many low- and middle-income countries; hence it is very meaningful. The manuscript was easy to read and follow, and it is accompanied with clear descriptions and graphs in most parts. However, there are some concerns for the authors to consider.

Major concerns:

1. The authors proposed a Bayesian approach where the denominators were assumed to follow a Normal model. However, Normal assumes a symmetric distribution of the data around the mean. Did the authors attempt a more flexible distribution such as a negative binomial? It seems more plausible for modelling counts data, too.

2. Directly truncating the denominator estimates would make the left-over Normal model have a higher mean and smaller variances compared to what the authors specified in the original inputs. Without a proper adjustment, this will cause the ratio estimate in the end to bias downwards because you were sampling your denominators from a distribution with its mean larger than what you expected. The authors should investigate the potential bias of their ratio estimators and discuss the influence of such.

3. Similar to point 2, a truncated Normal also inflates Δ_j1 and deflates Δ_j2 in the sensitivity analysis. The former case may be less of a concern but the latter one should be addressed appropriately.

4. In the estimation of denominator means through a simple linear regression, the vaccination tallies data at time t-1 were used as one commune factor. But the time t vaccination tallies were then used as the numerator, which leads to a concern of autocorrelation in the vaccination data.

5. Another maybe more intuitive specification is to directly model the conditional probability of polio vaccine coverages, that is, p_j |n_j,X_j ∼ Beta(α_j, β_j), where α_j and β_j can be estimated using the 19 communes similar to what the authors had done, and X_j’s are the two commune factors. In this case, n_j's can be modelled marginally using a Poisson or a simple linear regression, while taking into account the autocorrelation altogether.

Two potential advantages of doing it this way are: 1) it automatically requires the outcome ratio to be between 0 and 1 so it avoids the “truncation bias”; and 2) Beta allows more flexibility in the shape than Normal does. I would like to see how this approach performs compared to the authors' original one.

Minor concerns & suggestions:

1. There appears to be only 18 points on Figure 2, which commune was missing?

2. I did not fully understand how the authors came up with the variances in the normal model (page 9). Please provide more details.

3. I assume the first paragraph in the Results section deals with the method using direct regression estimates. The authors should be clearer about what they are writing here.

4. For Figures 3 and 4, are those "hybrid" grey dots the 19 communes used as the baseline? Also, it would be helpful to see the prediction intervals for those 19 communes as a proof of concept to make sure at least the model is interpolating properly.

5. Although the authors clarified their use of a simple linear regression in the Discussion section, it was originally unclear what method was used when I was reading the Methods on page 9 for the first time, as it only says they “built a regression model” with no clarification in terms of the regression model forms.

6. What if only the 2013 census data with adjustments of population growth rates was used as the commune factor? It may be interesting to include this as part of the sensitivity analysis to see how sensitive the estimators are to the “quality” of data that the denominator estimates are based upon. It is also possible that the population did not change as much between 2013 and 2015 so you would see almost identical results.

7. The figure labels were referenced incorrectly in Additional File 2.

Reviewer #2: This is an interesting paper, that address an important problem in global health metrics: how to obtain coverage indicators from health facility/health service data. The authors, in particular, grapple with a vexing issue: the frequent occurrence of coverage rates > 100% due to inaccurate denominators. They propose a solution that relies on combining multiple data sources, along with Bayesian methods. I thought the paper and methods were clear, but several revisions/clarifications would improve the manuscript.

1) how frequent are LQAS surveys? this is unclear from the manuscript, but the utility of the methods depends directly on the availability of such sample surveys. Are they conducted every year? less frequently? more frequently? we are only told that they are conducted widely, and the review cited covers the period 1984-2004. Maybe a map of LMICs indicating recent LQAS data availability could be inserted (e.g., time since the last LQAS survey in each country).

2) the ratio estimator depends critically on the accuracy of the parameter n, i.e., the number of children receiving a service or intervention. This is treated at some length in appendix 2, via sensitivity analyses, but I think at least one of the figures of that appendix should be moved to the main text, and discussed in the abstract. Otherwise, i think the reader might not accurately grasp the uncertainties that still underpin this method.

3) there are also studies in LMICs that have looked at the completeness of administrative via record linkages, for example, or re-tallying of facility registers. it might be worth citing these studies.

4) some of the features of the model (e.g., truncation by reality) could also be applied to outdated census data (and associated projections) that underpin current estimates of coverage. In fact, it would be quite interesting to see how close to the hybrid estimates one might get with only census data as input for d + an appropriate Bayesian framework. I'm asking because such projections from census data are available for every year-area, with uncertainty increasing with time since last census and smaller areas.

5) What is the inter-intervention of the denominator estimates? by this, i mean, that we obtained an estimate based on the coverage of vaccination, but do we get something similar based on, say, a nutrition intervention? this goes back to issues of completeness of the admin data, but it would be very interesting to learn if such an assessment is possible in Benin.

6) i suspect the sample of the Benin survey relied on census data for parts of the selection process (e.g., maps?). This is unclear to me from the text, or from the references cited. it would help to have that detail. If census data is used at any stage, how is the uncertainty it carries accounted for in the model?

6. PLOS authors have the option to publish the peer review history of their article (what does this mean?). If published, this will include your full peer review and any attached files.

**Do you want your identity to be public for this peer review?** For information about this choice, including consent withdrawal, please see our Privacy Policy.

Reviewer #1: No

Reviewer #2: No

---

## [Editor Report · Decision Letter 1]

6 Mar 2022

How to estimate health service coverage in 58 districts of Benin with no survey data: using hybrid estimation to fill the gaps

PGPH-D-21-00820R1

Dear Professor Valadez,

We are pleased to inform you that your manuscript 'How to estimate health service coverage in 58 districts of Benin with no survey data: using hybrid estimation to fill the gaps' has been provisionally accepted for publication in PLOS Global Public Health.

Best regards,

Karen A. Grépin, Ph.D.

Academic Editor

Dear Authors,

I would like to thank the research team for considering and putting in a great deal of effort to address the comments and concerns raised by both reviewers. I think the authors have done a sufficiently good job at addressing the concerns raised by the reviewers that I happy to now accept this paper for publication, subject to any necessary modification to satisfy journal publishing requirements.

Karen Grépin